# Antifungal, Antibacterial, and Cytotoxic Activities of Silver Nanoparticles Synthesized from Aqueous Extracts of Mace-Arils of *Myristica fragrans*

**DOI:** 10.3390/molecules26247709

**Published:** 2021-12-20

**Authors:** Humaira Rizwana, Najat A. Bokahri, Fatimah S. Alkhattaf, Gadah Albasher, Horiah A. Aldehaish

**Affiliations:** 1Department of Botany and Microbiology, College of Science, King Saud University, P.O. Box 22452, Riyadh 11495, Saudi Arabia; najatab@ksu.edu.sa (N.A.B.); falkhataf@ksu.edu.sa (F.S.A.); haldehish@ksu.edu.sa (H.A.A.); 2Department of Zoology, College of Science, King Saud University, Riyadh 11451, Saudi Arabia; Galbeshr3@gmail.com

**Keywords:** mace, *Myristica fragrans*, AgNPs, antifungal activity, antibacterial activity, cytotoxic activity

## Abstract

In the present study, mace-mediated silver nanoparticles (mace-AgNPs) were synthesized, characterized, and evaluated against an array of pathogenic microorganisms. Mace, the arils of *Myristica fragrans*, are a rich source of several bioactive compounds, including polyphenols and aromatic compounds. During nano synthesis, the bioactive compounds in mace aqueous extracts serve as excellent bio reductants, stabilizers, and capping agents. The UV-VIS spectroscopy of the synthesized NPs showed an intense and broad SPR absorption peak at 456 nm. Dynamic light scattering (DLS) analysis showed the size with a Z average of 50 nm, while transmission electron microscopy (TEM) studies depicted the round shape and small size of the NPs, which ranged between 5–28 nm. The peaks related to important functional groups, such as phenols, alcohols, carbonyl groups, amides, alkanes and alkenes, were obtained on a Fourier-transform infrared spectroscopy (FTIR) spectrum. The peak at 3 keV on the energy dispersive X-ray spectrum (EDX) validated the presence of silver (Ag). Mace-silver nanoparticles exhibited potent antifungal and antibacterial activity against several pathogenic microorganisms. Additionally, the synthesized mace-AgNPs displayed an excellent cytotoxic effect against the human cervical cancer cell line. The mace-AgNPs demonstrated robust antibacterial, antifungal, and cytotoxic activity, indicating that the mace-AgNPs might be used in the agrochemical industry, pharmaceutical industry, and biomedical applications. However, future studies to understand its mode of action are needed.

## 1. Introduction

Nanotechnology research and development, and its applications in agriculture and medicine, is a contemporary and ongoing process. This technology has been explored by various other sectors of bioscience with successful outputs. The success of nanoparticles lies in their miniature size, which allows them to permeate and reach the targeted cell with great ease. Globally, the past several years have shown a dramatic escalation in bacterial strains that are resistant to a number of antibiotics. As a result, the medical and pharmaceutical industries have been compelled to develop alternative, novel drugs to combat their spread and proliferation [1]. Similarly, the food and crop production sectors are in dire need of an alternative control method to fungicides, to combat tenacious and persistent fungal plant pathogens. The indiscriminate use of synthetic pesticides and fungicides has resulted in the emergence of several resistant fungal strains. As a result, the agri–food industry is conducting extensive research into alternative biologicals, including green synthesized nanomaterials. Huge losses in food commodities due to plant pathogenic fungi have always hampered the anticipated food supply and agricultural produce for the growing world population. The increasing demand for safe food supply has prompted scientists to explore and research healthier alternatives that impede the growth of phytopathogens, thus increasing agricultural yields and food production [2,3]. Furthermore, the nutritive value and quality of the food must be preserved [3].

Nanotechnology is an innovative technique that creates nanoparticles by employing several metals (zinc oxides, copper oxide, gold, and silver) [4,5]. Over the past few years, there has been a significant upsurge in the application of biologically synthesized nanoparticles (NPs) over nanoparticles synthesized through conventional methods, including chemical and physical methods. The biological method of synthesis is quicker, less expensive, and less toxic than other methods [5,6]. This method employs a variety of biological materials, including fungi [7], algae [8], bacteria [9], plants [10], and biological compounds [11]. Amongst them is the green synthesis of NPs, which uses plant materials and their extracts. Green synthesis of NPs is also known as phyto-fabrication, because it employs a variety of plant parts such as roots [12], stems [13], leaves [14], flowers [15], fruits [16,17], and seeds [18] for the fabrication, using different metals and their oxides. Previous research on the green synthesis of NPs has demonstrated that plant materials are very appropriate and reliable for the bio-fabrication of metallic NPs [19]. Singh and colleagues in 2010 [20] synthesized AgNPs from an aqueous leaf extract of *Argimone maxicana*. They reported the rapid synthesis of NPs that were crystalline in nature and highly stable, measuring 15–30 nm in dimension and exhibiting strong inhibitory activity against bacterial test isolates. An aqueous leaf extract of *Chenopodium album* yielded NPs with quasi-spherical shapes and sizes ranging from 10 to 30 nm [21]. According to a previous study, AgNPs synthesized from extracts of thyme leaves and ginger rhizomes exhibited strong antifungal activity against *Candida albicans*, when compared with the standard antifungal fluconazole [22]. Silver and gold nanoparticles were synthesized from seed and fruit extracts of *Artocarpus heterophyllus* Lam [18], *Cuscuta japonica* [23], *Illicium verum* [24], *Trigonella foenum-graecum* L [25], and *Emblica officinalis* [26]. In a previous study, AgNPs synthesized from aqueous seed extracts of Cuscuta japonica exhibited potent antibacterial activity. The strong bacterial inhibition was attributed to several bioactive compounds identified in the FTIR studies of both extracts, and AgNPs of *C. japonica*, which included esters, alcohols, phenols, and carboxylic acids. Similarly, *Artocarpus heterophyllus* Lam seed powder extracts were used to synthesize small AgNPs (3–25 nm) with very strong antibacterial activity against both gram-positive and gram-negative bacteria, including *Staphylococcus aureus*, *Bacillus subtilis*, and *Escherichia coli*. Phyto-fabrication is therefore regarded as an excellent and simple approach for the fabrication of NPs, as it has opened the path to more sustainable, cleaner, and safer nanoproducts [27].

Furthermore, green synthesis is a highly sought-after process in biological synthesis because it does not necessitate the use of growth culture media or special conditions to support and maintain the growth of biological organisms [6]. Besides, the raw material for green synthesis (plant material) is abundant in nature, and inexpensive. Their synthesis is relatively faster, resulting in pure, environmentally friendly, and safe nanoparticles with an enhanced performance. Plants, as enormous reservoirs of diverse chemical compounds, play a pivotal role in providing several drugs to the pharmaceutical industry, which are used as therapeutic agents for several illnesses. Plant extracts contain phytocompounds such as carbonyl compounds, alkaloids, flavonoids, terpenes, polyphenols, and several proteins that act as capping, reducing, and stabilizing agents during the synthesis of NPs [28,29].

Among all the metals, silver is the most widely used, especially in the fabrication of NPs. AgNPs are already established and are extensively used in diverse fields such as nano fertilizers [30], nano pesticides [31], nanomedicine [32], drug delivery [33], as anti-microbial products [34] and in several biomedical applications. In spite of a few reports that point out the adverse toxicity effects of AgNPs, they have received considerable approval as an antibacterial, antifungal agent, and disinfectant [35].

*Myristica fragrans* Houtt is an aromatic evergreen tropical plant and a member of the family Myristicaceae. The seed of this plant is commonly referred to as nutmeg, while the arils are known as mace. The arils appear as a network around the seed. Both the seed and the arils are commonly used spices in a variety of cuisines [36]. In many parts of Asia, mace is referred to as javitri or bisbasah. In Unani medicine, Ayurvedic, and Chinese traditional medicine, mace is commonly used for treating peptic ulcers, flatulence, stomach discomfort, gastrointestinal symptoms, and anxiety [36,37,38]. Very few reports have shown the antibacterial activity and anti-inflammatory properties of mace extracts. However, to our knowledge, there are no reports on the antifungal and antibacterial activity of silver nanoparticles synthesized from mace aqueous extracts. Hence, the present study aimed to green-synthesize silver nanoparticles using the arils of M. fragrans and evaluate their cytotoxic and antimicrobial activity against a wide array of pathogenic microorganisms.

## 2. Results and Discussions

### 2.1. Synthesis of Mace-AgNPs from Aqueous Extracts of Mace (Arils of Myristica Fragrans)

The formation of AgNPs from plant material is a quick and productive method that can be easily implemented in a standard laboratory setting with minimal effort. An aqueous extract was prepared by adding crushed reddish-brown arils of Myristica fragrans (mace) to distilled water. The mixture was boiled for 20 min and left at room temperature for 24 h. After that, it was filtered, centrifuged, and the supernatant was used for the preparation of mace-AgNPs.

To create silver nanoparticles, the aqueous extract of the arils (mace) was added to a colorless solution of AgNO3, which resulted in a yellowish reaction mixture. The reaction mixture was exposed to sunlight. After 24 min in direct sunlight, the mixture started to change its color to a pale, brownish solution. The entire nucleation process, as indicated by the deep brown color caused by the formation of AgNPs, took 30 min (Figure 1). Aside from the visual examination, the formation of mace-AgNPs was authenticated by the absorption spectrum of AgNPs obtained by the UV-VIS spectrophotometer. An LSPR band was clearly visible after 26 min. The band intensity increased until 30 min and then settled with no further changes, indicating complete reduction of Ag+ and the completion of green synthesis (Figure 2). The color change and subsequent LSPR band at 456 nm show the harmonious oscillations of electrons in silver nitrate in resonance with light waves, resulting in an SPR band [39,40]. Similarly, LSPR absorption peaks between 420 and 478 nm have been reported earlier in the AgNPs synthesized from nutmeg (seeds of *Myristica fragrans*) [41,42].

The rapid color change of the reaction mixture mediated by sunlight and the subsequent development of an absorption peak at 456 nm in the UV-Vis spectrum clearly indicated the formation of mace-AgNPs. Similar to our findings, rapid sunlight-mediated synthesis of AgNPs from *Polygonatum graminifolium* (30 min) and *Sida retusa* leaf extracts (60 min) was reported in earlier studies [43,44]. A previous study had proposed a comprehensive mechanism for the role of UV light and the influence of electromagnetic irradiation on the creation or synthesis of AgNPs [45]. Another point of view is that UV light from the sun cannot pass through the glass tubes and instruments used in the synthesis of NPs [46]. Hence, the blue light of the visible spectrum could play a major role during the reduction of Ag+ to Ag [46,47]. Recently, Nguyen and colleagues in 2020 [48] proposed that the blue light causes the flavonoids in the plant extracts to convert from enol to keto form (tautomerize), resulting in the release of hydrogen atoms, which could possibly influence the reduction process of Ag+ ions.

### 2.2. Dynamic Light Scattering Analysis (DLS)

A Zeta sizer (Zeta sizer-ZEN-3600) was employed to establish the size distribution of the synthesized Mace-AgNPs. The DLS spectrum depicts the size distribution of nanoparticles in a hydrodynamic state, which is intensity weighted (Figure 3). The mace-AgNPs had an average size of 50.70 nm and a polydiversity index (PDI) of 0.220. A lower PDI value, less than 0.7, indicates good quality of synthesized nanoparticles with a narrow size distribution range [49]. The coating of plant extract around synthesized nanoparticles influences their hydrodynamic diameter [50]. Green nanoparticles in the same range with a low PDI value have previously been reported [51].

### 2.3. Transmission Electron Microscopy Study of the Synthesized Mace-AgNPs

The DLS spectrum gives precisely the size and distribution of NPs in aqueous solutions, but TEM provides a clear picture of the morphological characteristics of NPs, such as their size and shape. The microphotographs depict the nanoparticles as being well separated and spherical (Figure 4). Figure 4 also shows a few spheroidal NPs. The particle size ranged from 5 to 28 nm. Similar to our findings, a previous report on AgNPs synthesized from nutmeg measured between 5 and 20 nm [52]. Small-sized nanoparticles are indicative of excellent biological activity [53]. The variation in the size of the mace-AgNPs between DLS and TEM is because the NPs are measured in a dry state in TEM, whereas in DLS they are measured in a hydrodynamic state, which includes the size of the biomolecules, including the organic surface coatings that are made up of the tightly adhering solvent molecules and the metal core [54]. Furthermore, the DLS observations rely heavily on Rayleigh scattering caused by NPs suspended in fluids [55]. Additionally, the larger size of NPs in DLS measurements could be due to the influence of Brownian movements as the NPs are dispersed in liquid [56].

### 2.4. The Elemental Analysis of Synthesized Mace-AgNPs (FESEM-EDS)

The elemental analysis of synthesized mace-AgNPs showed several peaks on the EDS spectrum, including the typical peak of silver, as shown in Figure 5. At 3 keV, an intense LSPR absorption peak very peculiar to silver was clearly visible, and a weaker silver peak at 2.7 keV was also seen in the spectrum. The amount of nano silver in the sample was estimated to be 47.7% (wt%). The peak at 3 keV arising from silver suggests the synthesis of AgNPs [57]. Other elements recorded and clearly visible on the spectrum included: Si (15%); Zr (16%); S (6%); K (4%); Cl (3%), Zn (3%); Na (1%); Al (0.8%), and Ti (1.7 percent). Some elements, such as S, K, Na, Cl, and Al, may have been present in the mace extract, and become lodged in the AgNPs during nano synthesis. Similar to the present study, a recent study showed an intense peak at 3 keV in the EDX spectrum of AgNPs prepared from nutmeg extracts. In addition, they reported elements such as Na, Zn, K, and Ca in the elemental profile and referred to these as micronutrients or biomolecules of the nutmeg extract [41].

### 2.5. The FTIR Spectrum of Mace Extract and Synthesized Mace-AgNPs

The FTIR spectrum of both the mace extract and the mace-AgNPs exhibited several peaks that are characteristics of important bioactive compounds, indicating the presence of phenols, alcohols, aromatic compounds, alkaloids, amines, and carboxylic acids (Figure 6 and Figure 7). The FTIR spectrum of mace aqueous extract showed peaks at 3410 cm^−1^, 2930 cm^−1^, 2366 cm^−1^, 1622 cm^−1^, 1415 cm^−1^, 1239 cm^−1^, and 1075 cm^−1^, and peaks in the fingerprint region (Figure 6). However, the IR spectrum of mace-AgNPs showed slight shifts in peak positions. The peaks were observed at 3753 cm^−1^, 3428 cm^−1^, 2930 cm^−1^, 2369 cm^−1^, 1632 cm^−1^, 1388 cm^−1^, and 1075 cm^−1^ (Figure 7). Strong and sharp peaks between 3400 and 3753 cm^−1^, and a weak broad peak at 2930 cm^−1^ in both the extracts, and synthesized mace-AgNPs, indicated the presence of alcohols and phenols arising from OH stretching vibrations due to the CH asymmetric stretching of alkanes. The peaks at 1415 cm^−1^, 1239 cm^−1^, and 1075 cm^−1^ corresponded with the C-N skeletal stretching vibrations of amines. The weak and sharp peaks at 1622 cm^−1^ and 1632 cm^−1^ corresponded with the carbonyl stretch of amides in proteins. Similar to our findings, previous reports have shown the presence of carbonyl groups, alcohols, phenols, and alkene-related peaks in the FTIR spectrum of mace, nutmeg extract, and synthesized NPs [42,52,58]. The shifts in the positions of bands and the disappearance of some peaks in the IR spectrum of synthesized mace-AgNPs indicated the role of OH (hydroxyl) phenols, alcohols, and the carbonyl group of amines in the capping and reduction of silver ions to AgNPs. Carbonyl groups in proteins and hydroxyl groups in alcohols have a strong ability to bind to metals, implying the formation of a layer around the metal NPs [42,59], resulting in capping and preventing the agglomeration of NPs [59,60].

Mace extracts and synthesized copper NPs showed the presence of aromatic rings, esters, phenols, and carboxylic acids [61]. The aforementioned bioactive compounds found in mace are typical biomolecule markers present in the FTIR spectrum of green synthesized nanoparticles [62]. Very similar to our findings, these biomolecules were reported earlier from *M. fragrans* arils, seed extracts, AgNPs and essential oils [42,63,64]. Sasidharan and colleagues, in 2020 [41], synthesized silver and copper NPS from the pericarp of *M. fragrans* fruits. They identified identical compounds in FTIR studies as reported in the present study, and suggested the involvement of eugenol, phenolic compounds, and fatty acids in the synthesis of nanoparticles.

### 2.6. Antifungal Activity of Mace Aqueous Extracts and Synthesized AgNPs

The antifungal activity of mace aqueous extracts and synthesized AgNPs was assessed against several phytopathogenic fungi. Figure 8 clearly depicts the significant mycelial growth inhibition of *A. alternata*, *F. oxysporum*, and *P. magniferae* when treated with extracts and AgNPs. However, *T. harzianum* exhibited poor inhibition by both the extract and the synthesized AgNPs. It was observed that the AgNPs inhibited the growth and proliferation of mycelium more strongly than the extracts. The extracts caused inhibition of mycelial growth in a variable manner. Figure 9 shows the diameter of the fungal colonies; consequently, Figure 10 depicts the percentage mycelial inhibition of test fungi, which are as follows: *P. magniferae* (50%), followed by *A. alternata* (13.70%) and *F. oxysporum* (23.75%). In comparison, the AgNPs were more effective as the inhibition was stronger. *P. magniferae* demonstrated the highest radial growth inhibition, when treated with mace-AgNPs (89%), followed by *A. alternata* (83%) and *F. oxysporum* (76%). There is very little data available on the antifungal activity of green-synthesized NPs. To our knowledge, there are no reports that demonstrate the antifungal activity of AgNPs against phytopathogenic fungi. However, a recent study by Fernando and Senevirathne showed the complete inhibition of the growth of *F. oxysporum* when treated with organic extracts of mace [65]. Yet, another study demonstrated the significant antifungal activity of three lignans isolated from nutmeg. The lignans showed the significant inhibition of a wide variety of plant pathogenic fungi that cause tomato blight, wheat rut, barley powdery mildews, and rice blast [66]. The excellent antimicrobial properties shown by nutmeg seeds could be due to compounds such as carvacrol, b-caryophyllene, p-cymene, and α-pinene, present in the nutmeg seeds [67].

### 2.7. Antibacterial Activity of Aqueous Extracts and AgNPs Synthesized from Mace

The antibacterial activity of mace extracts and AgNPs is depicted in Figure 11 and Figure 12. The antibacterial study showed that mace-AgNPs were highly effective in inhibiting the bacterial test isolates, whereas the mace aqueous extract was the least inhibitory, as none of the test isolates showed inhibitory zones, indicating the poor inhibitory activity of mace extracts. *S. aureus* showed the biggest zone of inhibition (22 ± 1.25) when treated with mace-AgNPs, followed by *E. coli* (20 ± 0.75) and *B. subtilis* (19 ± 1.58), respectively. AgNO_3_ also showed weak inhibition of some *B. subtilis* and *E. coli.* Similar to the findings of this study, previous research has shown that aqueous extracts of nutmeg did not inhibit the bacterial test isolates, *S. aureus* and *E. coli*, but the synthesized NPs showed significant growth inhibition of the aforementioned bacterial species [52]. Recent studies have shown the excellent antibacterial activity of silver AgNPs and CuNPs, synthesized from the seeds of *M. fragrans* against *S. aureus*, *E. coli*, *P. aeruginosa*, *B. subtilis*, and *Salmonella typhi* [41,42,52]. Gram-negative bacteria are more susceptible to Ag ion penetration than gram-positive bacteria due to the differences in cell wall composition and structural organization [68]. However, in the present study, *P. aeruginosa* exhibited poor antibacterial activity. Similar to our findings, in a previous study, hospital strains of *P. aeruginosa* showed resistance towards AgNPs and tested antibiotics [69]. The possible reasons for the resistance shown could be the result of the decreased permeability of the cell membrane, alterations in the structure of porins, or the activity of efflux pumps [69].

As early as 1987, Woo and colleagues [70] isolated two lignans (Macelignan and Meso-dihydroguaiaretic acid) for the first time from mace. Lignans isolated from mace and seeds of *M. fragrans* showed strong antifungal activity against several plant pathogenic fungi, including *Agrobacterium tumefaciens*, *Alternaria alternata*, *Colletotrichum gloeosporioides*, *C. coccodes*, and *Magnaporthe grisea* [66]. Similarly, dihydroguaiaretic acid from mace arils exhibited strong inhibition of *Helicobacter pylori* at a low MIC of 100 µg/mL [71]. Another study isolated and identified two compounds, malabaricone B and malabaricone C, from mace extract [72]. These compounds were found to have potent antifungal and antibacterial properties against *Staphylococcus aureus*, *Bacillus subtilis*, and *Candida albicans* [72]. Previous research has confirmed the presence of neolignans, lignans, phenylpropanoids, flavonoids, esters, and phenolic constituents [73,74,75,76,77] in the aqueous and organic solvent extracts of mace. Mace oils have also shown the presence of several terpenes, and its derivatives [78]. All the compounds referenced above have demonstrated a broad spectrum of antimicrobial activity. Mace licarins caused the complete growth arrest of bacteria that cause dental caries, i.e., *Streptococcus mutans* [79]. Essential oils extracted from mace showed the presence of quite a few monoterpene hydrocarbons and oxygenated monoterpenes, that included limonene, -pinene, -terpinene, -pinene, linalool, terpinene-4-ol, and α-terpineol. The mace oils also demonstrated strong antibacterial and antifungal activity, which was attributed to their terpene composition [80].

The mode of action of AgNPs against microorganisms still needs a deeper understanding. However, several researchers have documented their findings in this regard, and have stated that the most probable reason for the robust inhibitory activity of AgNPs against both fungi and bacteria could be their competence to physically integrate with the cell membrane [81,82]. The most probable mechanism of antimicrobial activity is still debatable, as some researchers attribute it to AgNPs while others credit it to silver ions (Ag+). It was proposed that the AgNPs make direct contact with the cell wall, adhere strongly, and finally penetrate the microbial cell [82]. Close contact of AgNPs with the bacterial cell wall and their penetration into the cell cytoplasm impairs the cell membrane, which results in cell leakage, leading to cell death [83,84]. Additionally, once in the cytoplasm, the AgNPs interact with several molecules in the cell, similar to lipids, DNA, and proteins. The interaction of NPs with the respiratory enzyme system results in the generation of ROS and free radicals that induce oxidative stress, and interfere with vital metabolic functions by damaging nucleic acids and cell proteins [85]. Reports also suggest that the free radicals along with the NPs cause substantial damage to the cell membrane by forming several pores, disturbing the integrity of the cell membrane, and resulting in disintegration [67]. Another view is that the penetration of NPs results in the disruption of cell integration, inhibits vital processes including respiration, protein synthesis, and ion transport, and finally leads to cell death [86,87]. Previous studies have reported that NPs disrupt the ergosterol of the fungal cell membrane, disturb the osmotic balance through downregulation of oxidative enzymes, and the generation of ROS [88]. Such disturbances often cause cell instability and ultimately culminate in cell death [89]. Another study reported that AgNPs disturbs the lipid bilayer of the cell membrane due to altered permeability, resulting in the leakage of cell materials and cell death [90].

Contrary to the aforementioned view, the antibacterial activity of AgNPs is believed to be due to the silver ions, which are released continuously from AgNPs [91]. Metals oxidize in aqueous solutions to release metal ions. Hence, AgNPs yield Ag+ ions in aqueous solutions [92,93]. Due to the affinity and electrostatic attraction of silver ions towards sulphur proteins that are bound to the cell wall, the silver ions strongly adhere to the cell wall and cell membrane, thereby enhancing the permeability of the cell membrane and damaging the cell envelope [93,94]. Their entry into the cytoplasm deactivates the respiratory enzymes and generates ROS (reactive oxygen species), which incites the disruption of the cell membrane and inhibits ATP production and protein synthesis, besides damaging the DNA by interacting with phosphorus and sulphur [94,95], ultimately leading to cell death. Another view is that the antibacterial activity is the result of a synergistic effect of both the proposed antibacterial mechanisms. The direct contact and adherence of AgNPs to the cell membrane stimulates the release of Ag+ ions and their uptake by cells [96].

Hence, the potent inhibitory activity of mace-AgNPs on the growth of bacterial and fungal isolates could be accredited to the very small dimensions of the AgNPs and the bioactive molecules like carbonyl compounds, esters, and phenols attached to them. Previous findings have documented mace as a rich source of phenolic compounds, ester, aromatic rings, alkanes, methyl ester, alkenes, carbonyl groups, and aldehydes [58,61]

### 2.8. Cytotoxic Studies of Mace-AgNPs against the HeLa Cancer Cell Line

Cytotoxicity assessment with the sequential dilution of mace-AgNPs (3.125–100 µg/mL) was tested against the HeLa cell line and a dose-dependent curve was plotted, as depicted in Figure 13. The figure clearly displays that the viable cells markedly decreased as the concentration of mace-AgNPs increased, with an IC_50_ value of 18.05 ± 0.97 mL/100 mL. The significant cytotoxic activity of mace-AgNPs could possibly be a result of the synergy of the bioactive compounds (flavonoids, phenols, and amides) with the AgNPs and their ability to penetrate the cells, disrupt the ETS function, alter cell permeability, and cause cell death [97]. Secondary metabolites, including phenols and flavonoids, inhibit transcription and activate the generation of ROS, resulting in a surge of oxygen free radicals [98]. Consequently, oxidative stress increases, which is supposedly designated as the primary toxicity method of silver nanoparticles against cancer cells. The significant outcomes of the cytotoxicity of mace-AgNPs in this study are in complete concurrence with a previous finding that validates the cytotoxic activity of green-synthesized AgNPs against the cervical cancer cell line HeLa [99,100,101]. Recent research has added to the knowledge of the therapeutic use of green-synthesized AgNPs, but their mode of action and effects on normal cells still needs to be addressed.

## 3. Materials and Methods

### 3.1. Chemical Reagents

Various chemicals, reagents, and culture media were used to prepare the solutions and extract. All of them were of analytical grade and pure (99% purity). Silver nitrate, Potato dextrose agar and Nutrient agar were purchased from either Sigma Aldrich-Merck KGaA, Darmstadt, Germany or Thermo Fischer Scientific-Inc., Waltham, MA, USA.

### 3.2. Microorganisms

Four bacterial isolates and four fungal isolates were chosen for antimicrobial activity. The bacterial isolates (*Pseudomonas aeruginosa*, *Bacillus subtilis*, *Escherichia coli*, and *Staphylococcus aureus*) were kindly provided by the university hospital, the King Khaled hospital in Riyadh, Saudi Arabia. The fungal isolates (*Alternaria alternata*, *Pestalotiopsis mangiferae*, *Fusarium oxysporum*, and *Trichoderma harzianum*) were procured from the Department of Plant Protection, College of Food and Agricultural Sciences, King Saud University, Riyadh, Saudi Arabia.

### 3.3. Plant Material and Aqueous Extract Preparation

Mace was purchased from a local Unani store in Hyderabad, Telangana, India. Mace arils were roughly hand-crushed and employed for the aqueous extract preparation, as mentioned earlier [14]. Concisely, the crushed arils (10 g) were added to 100 mL of distilled water and boiled at 80 °C for 20 min. Then, the mixture was left for 24 h and allowed to cool at room temperature. After 24 h, it was filtered through Whatman’s filter paper (No. 1). The filtrate was then centrifuged at 5000 rpm (10 min), and the supernatant was collected to be employed for the preparation of silver nanoparticles, and for other experimental studies.

### 3.4. Synthesis of AgNPs Using Aqueous Mace Extracts

A solution of 1 mM silver nitrate (AgNO_3_) was prepared by mixing a fixed amount of silver nitrate powder to distil water. A reaction mixture of AgNO_3_ and mace extract was prepared by adding 45 mL of freshly prepared AgNO_3_ solution to 5 mL of aqueous mace extract. The reaction mixture changed to a brown from the initial buff color in 21 min. The color change indicated the formation of mace-AgNPs through reduction of Ag ions [85].

### 3.5. Characterization

UV-VIS spectroscopic analysis of the mace-AgNPs was conducted with an ultraviolet visible spectrophotometer (UV-VIS-Shimadzu, Japan-model No-1800). The dynamic light scattering analyzer (DLS) Model-Nano Series-Zeta sizer-ZEN-3600, Malvern, UK, was employed to ascertain the hydrodynamic size, polydiversity index (PDI), and the distribution of the synthesized nanoparticles. The mace nanoparticles were further screened for their elemental composition and the EDX spectrum was obtained. The field emission scanning electron microscope model JSM-7610F-Japan with an energy dispersive X-ray detector (EDS) was used to screen the elements in the nanoparticles and the spectrum was collected at an accelerating voltage of 30 kV. Furthermore, the shape and size of the mace-AgNPs were captured on a transmission electron microscope (TEM-JEOL JEM-Plus-1400, Tokyo, Japan). Preparation of all the samples for the analysis purpose was performed according to the manufacturer’s instructions. The synthesized nanoparticles were further screened with a Fourier transform infrared spectroscope in the scan range of 400–4000 cm (Thermo Scientific, USA, Model-Nicolet-6700) using a KBr pellet. Mace extracts were also subjected to FTIR analysis.

### 3.6. Antibacterial Activity

The antibacterial activity of the mace extract, mace-AgNPs, and AgNO_3_ was evaluated against the aforementioned bacterial isolates by following the agar well-diffusion method [102]. All the bacterial isolates were cultured separately in nutrient broth for 24 h to obtain a bacterial suspension corresponding to 0.5 Mac Farland suspension (∼10^6^ per mL-CFU-colony-forming unit). One hundred microliters (100 µL) of bacterial suspension were added to a solidified nutrient agar plate, and gently spread on the surface of the nutrient agar. After an hour, three wells with a diameter of 4 mm were punched in each plate. The wells were filled with 100 µL of either mace extract, mace-AgNPs, or AgNO_3_. All the bacterial isolates were screened in the similar manner mentioned above, and were incubated for a day (24 h) at 37 °C. Separately, for positive control, all the bacterial test isolates were screened in a similar manner against antibiotic discs (Tetracycline 30 µg). The antibacterial inhibitory activity was evaluated by measuring the clear zone around each well, which indicated the absence of growth. The diameter of each inhibition zone was measured (mm) and documented. The assay was carried out in triplicates.

### 3.7. Antifungal Activity

Pure cultures of plant pathogenic fungi were obtained by subculturing them for 7 days. A 6-mm disc of the fungal colony was removed with a sterile cork borer, and transferred to a PDA plate that had been amended with 500 µL of either mace extract, mace-AgNPs, or AgNO_3_. The PDA plates were modified by first adding a fixed amount of the test solutions (mace extract, mace-AgNPs, or AgNO_3_) to a sterile petri plate, followed by the addition of the molten PDA. The contents were thoroughly mixed and then allowed to solidify. Finally, the inoculated petri plates were incubated for 7 days at 28 °C. Mycelial growth was measured after 9 days, and the percentage mycelial inhibition was calculated [103]. The fungicides Mancozeb and Carbendazim (0.1%) were used as a positive control. All the fungal isolates were screened in a similar manner in triplicates.

### 3.8. Cytotoxic Effect of Mace-AgNPs on the HeLa Cancer Cell Line

Different concentrations of mace-AgNPs were tested for their in vitro cytotoxic activity against the HeLa cancer cell line by using the MTT assay. The HeLa cancer cell line (human cervical cancer cell line) was obtained from the American Type Culture Collection (USA). Concisely, to a 96-well plate, HeLa cells were added at a density of 1 × 10^4^ cells/well in 90 μL of DMEM (Dulbecco’s Modified Eagle’s Medium). After 24 h, the cells were treated with Mace-AgNPs (3.125–100 μg mL^−1^). The various concentrations were prepared by two-fold dilutions. The treated cells in the 96-well plate were further incubated for 24 h. After 24 h, MTT was added, and the plates were incubated for another 4 h. The treated plates were incubated for 24 h and MTT was added, after which the plates were incubated for another 4 h. Cell viability was determined by a colorimetric method and the percentage of cells viable was calculated with a microplate reader (SunRise, TECAN, Inc., San Bruno, CA, USA). The cytotoxic effect of standard drugs was also determined. The negative control (untreated) showed 100% of the cells viable. The IC _50_ value was assessed by plotting a dose-dependent curve, plotted using Graph Pad Prism (version 7-San Diego, CA, USA) [103].

### 3.9. Statistical Analysis

The data and values presented in the present study were analyzed by standard deviation, and analysis of variance (ANOVA) and Tukey’s HSD test were implemented for significant differences (*p* ≤ 0.05). The statistical tests were run on XLSTAT (software version 1 January 2020) and Graph pad prism version-8.4.3.686.

## 4. Conclusions

Green nanotechnology is an excellent, environmentally friendly approach to the synthesis of NPs using plants and other biological sources. This process, besides being cost-effective, is associated with minimal toxicity. Mace arils are a rich source of phenols, safrole, myristic acid, minerals, bioactive compounds, myristicin, and antioxidants. The present study demonstrated the significant reducing and stabilizing potential of biomolecules belonging to phenols and carbonyl groups that assisted in the nucleation and formation of small-sized AgNPs. The potent antibacterial, antifungal, and cytotoxic activity of mace-AgNPs in the present study could be ascribed to the small size and spherical shape of NPs that measure between 5–28 nm, as observed in TEM studies. The strong antimicrobial and cytotoxic activities suggest that the mace-AgNPs have good scope to be used in a variety of applications, including the agrochemical sector, pharmaceutical industry, and several biomedical applications. However, studies to establish the toxicity effects on cells are required.

## Figures and Tables

**Figure 1 molecules-26-07709-f001:**
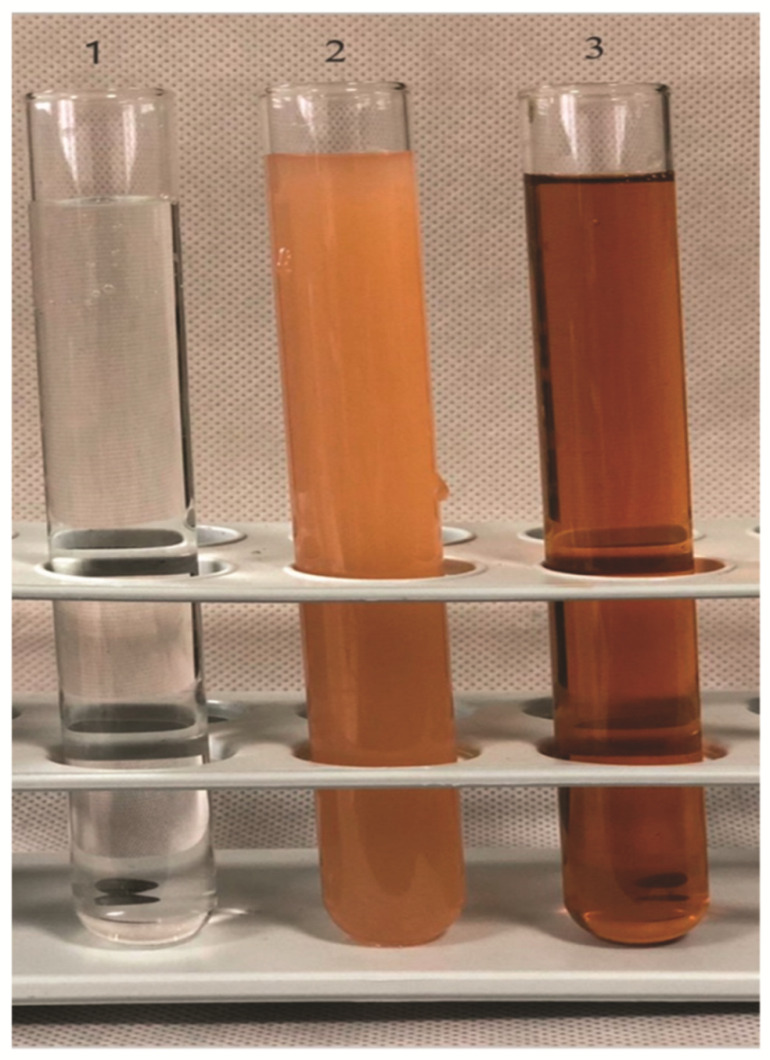
The production of mace-AgNPs using mace aqueous extracts. 1—silver nitrate, (AgNO_3_); 2—aqueous mace extract; 3—the colloidal solution of synthesized mace-AgNPs.

**Figure 2 molecules-26-07709-f002:**
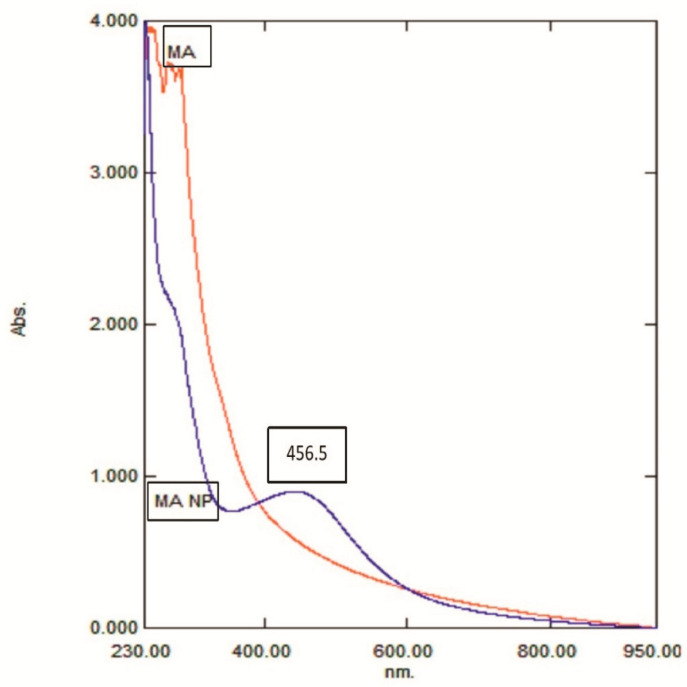
UV-VIS spectrum showing the LSPR peak of synthesized mace-AgNPs at 456 nm. MA-mace aqueous extract, MA NP-mace AgNPs.

**Figure 3 molecules-26-07709-f003:**
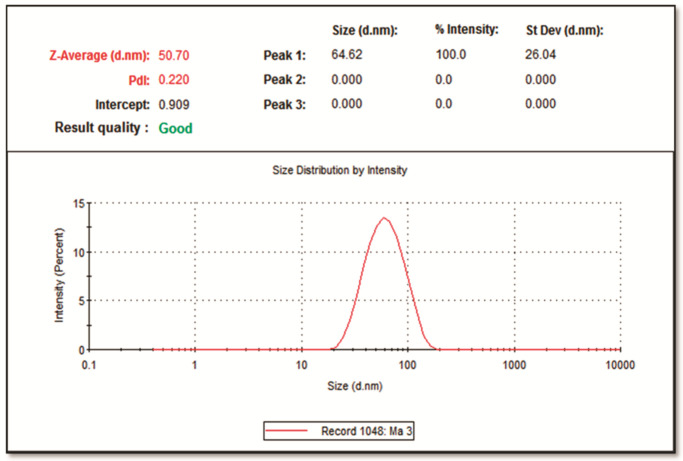
Dynamic light scattering (DLS) measurement of the average size and the size distribution of synthesized mace-AgNPs.

**Figure 4 molecules-26-07709-f004:**
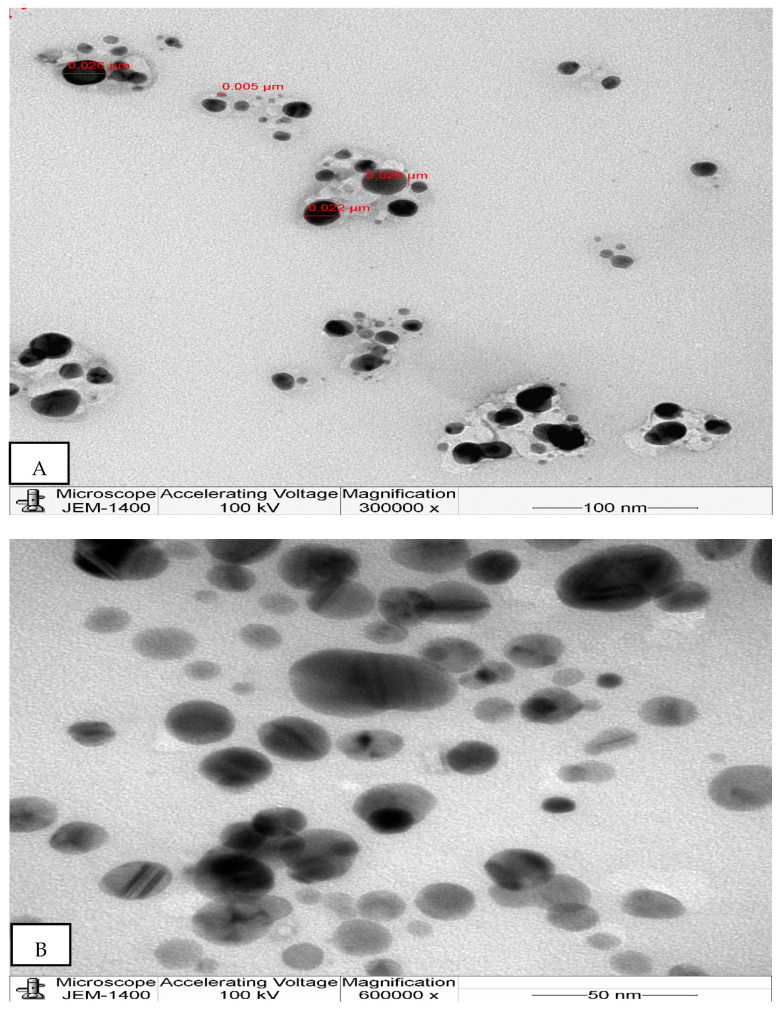
(**A**,**B**) Transmission electron microphotograph showing the size and morphology of the synthesized mace-AgNPs.

**Figure 5 molecules-26-07709-f005:**
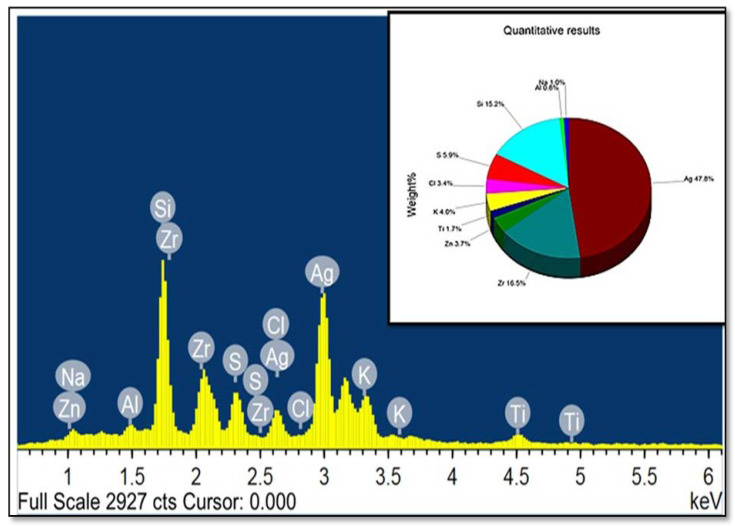
Energy dispersive X-ray spectrum of mace-AgNPs showing a prominent absorption peak of silver at 3 KeV.

**Figure 6 molecules-26-07709-f006:**
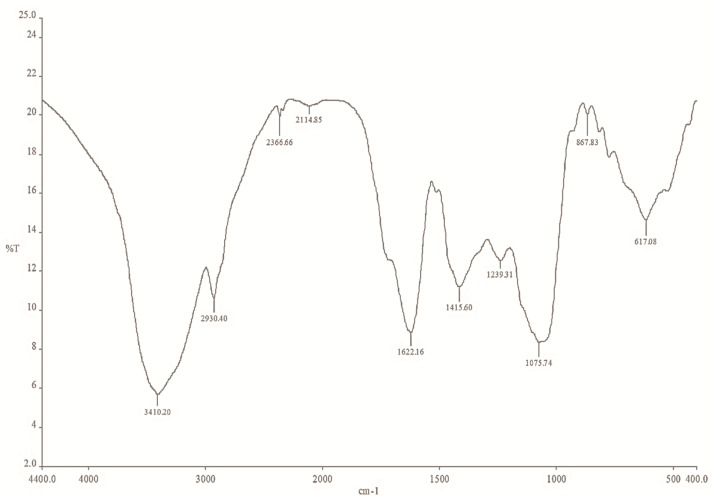
FTIR spectrum of mace aqueous extracts showing peaks of bioactive functional groups obtained on a Nicolet Spectrometer, in the range of 500–4000/cm^−1^.

**Figure 7 molecules-26-07709-f007:**
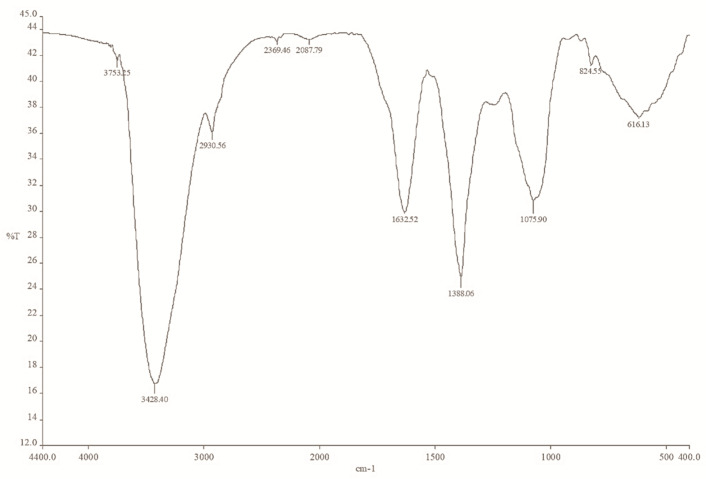
FTIR spectrum of mace-AgNPs showing peaks of bioactive functional groups obtained on a Nicolet Spectrometer, in the range of 500–4000/cm^−1^.

**Figure 8 molecules-26-07709-f008:**
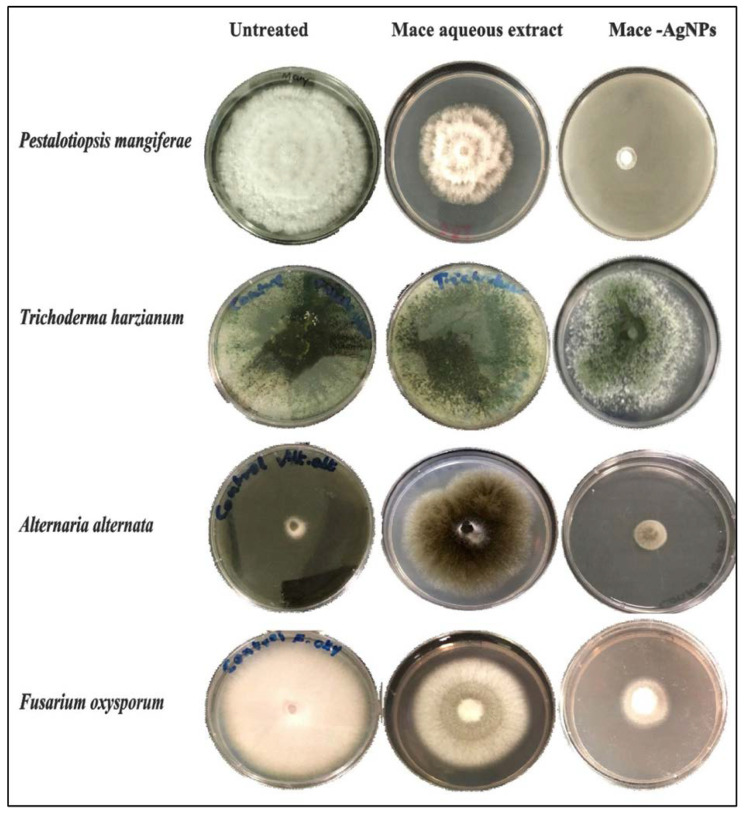
Antifungal activity of the mace aqueous extracts, mace-AgNPs and AgNO_3_ against plant pathogenic fungi.

**Figure 9 molecules-26-07709-f009:**
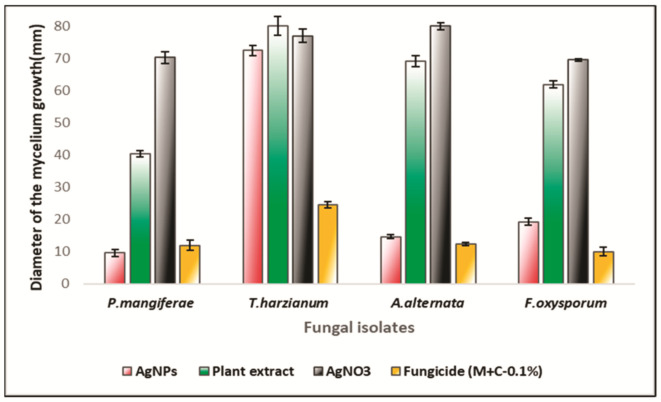
Diameter of the mycelial growth of fungal isolates treated with mace aqueous extracts, mace-AgNPs, AgNO_3_ and the fungicide (M+C). All values shown in the graph are means of three independent experimental replicates (±SD). Significant difference in means (*p* ≥ 0.05) were determined by analysis of variance (ANOVA) and Tukey’s HSD.

**Figure 10 molecules-26-07709-f010:**
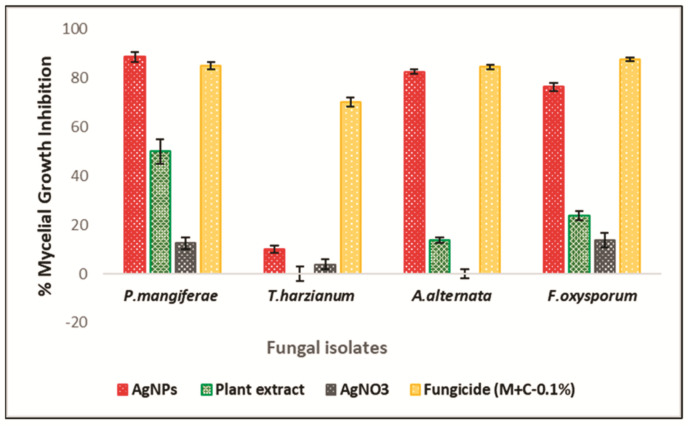
Percentage mycelial growth inhibition of fungal isolates treated with mace aqueous extracts, mace-AgNPs, AgNO_3_ and the fungicide (M+C). All values shown in the graph are means of three independent experimental replicates (±SD). Significant difference in means (*p* ≥ 0.05) were determined by analysis of variance (ANOVA) and Tukey’s HSD.

**Figure 11 molecules-26-07709-f011:**
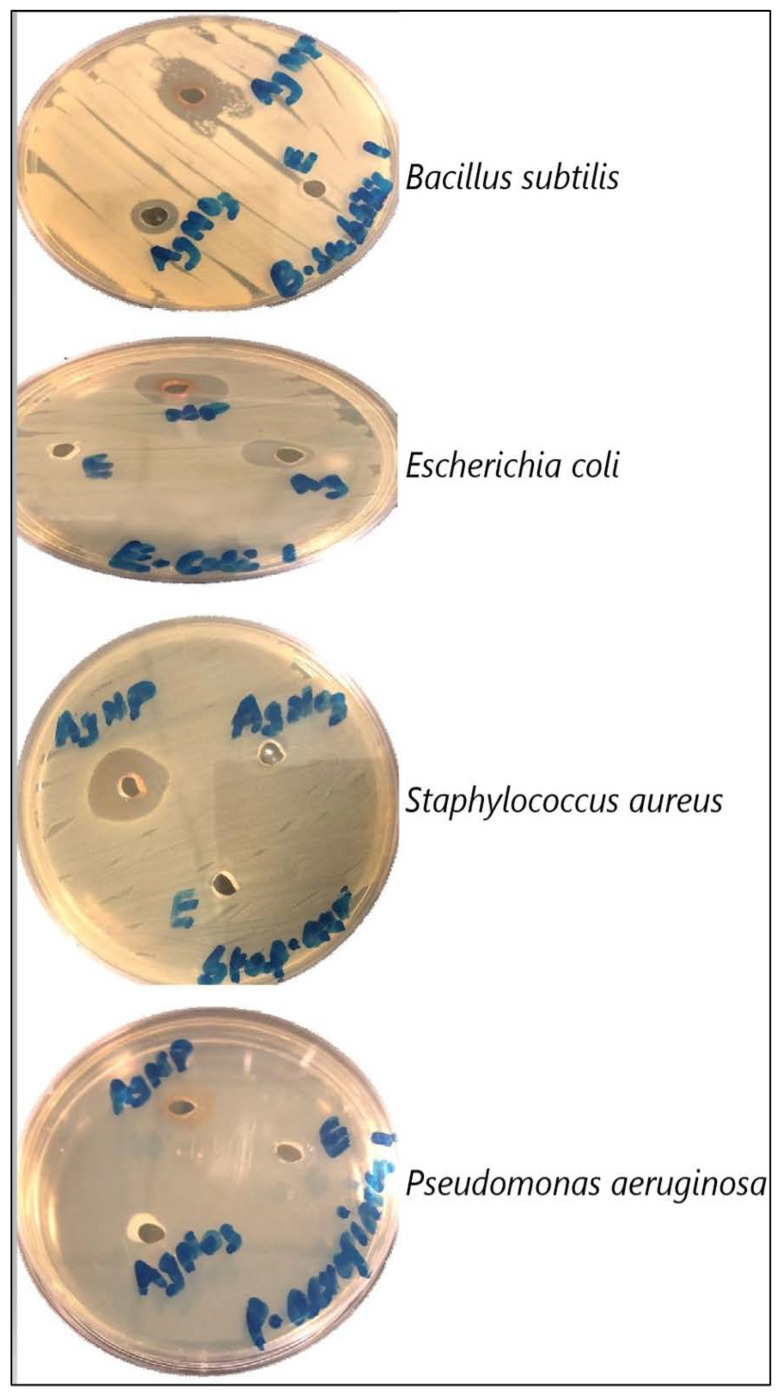
The antibacterial activity of mace-AgNPs, AgNO_3_ and aqueous mace extracts against pathogenic bacterial isolates.

**Figure 12 molecules-26-07709-f012:**
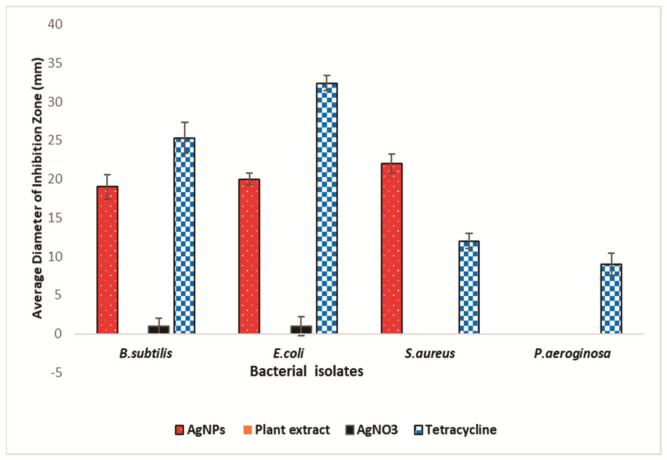
The antibacterial activities of mace extracts, synthesized silver nanoparticles, AgNO_3_ and Tetracycline were tested against Gram-positive and -negative bacterial isolates. Agar well-diffusion method was used to measure the clear zone of inhibition (mm) around each well. Three replicates were run for each set and the values were average (SD±). Significant difference in means (*p* ≥ 0.05) were determined by analysis of variance (ANOVA) and Tukey’s HSD test.

**Figure 13 molecules-26-07709-f013:**
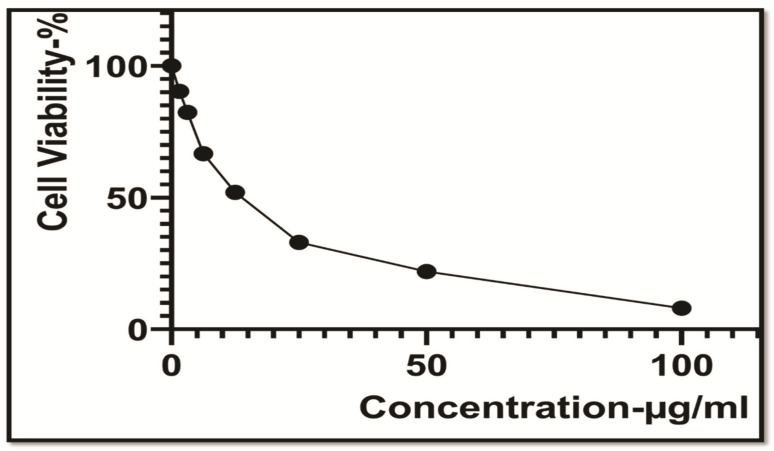
In vitro cytotoxic activity of mace-Ag-NPs synthesized using mace aqueous extracts and tested against HeLa cell line (cervical cancer cell line).

## Data Availability

All the data of the present study is present in the manuscript.

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
