# Peer review of "Antifungal, Antibacterial, and Cytotoxic Activities of Silver Nanoparticles Synthesized from Aqueous Extracts of Mace-Arils of Myristica fragrans"

_molecules, 2021, doi:10.3390/molecules26247709_

Round 1

Reviewer 1 Report

This work by Rizwana et al describe -the synthesis and characterization of silver nanoparticles by mace extracts and -their application in inhibiting the growth of certain bacteria, fungi and cancer cell lines. The article is well-presented and demonstrates applications of ‘green’ NPs, fitting well with the scope of MDPI molecules. Though I have (few major) suggestions/criticisms, which I recommend the authors to address.

  1. Line 57: Green synthesis of NPs is also termed as phyto-fabrication. Possibly a wider literature review could have helped identify some key works in the field such as (i) Phytofabrication of silver nanoparticles by leaf extract of Datura metel: hypothetical mechanism involved in synthesis. Journal of Bionanoscience, 3(1), pp.39-44, (ii) Phytofabrication of nanoparticles through plant as nanofactories. Advances in Natural Sciences: Nanoscience and Nanotechnology, 5(4), p.043002. and (iii) Removal of hexavalent chromium ions by Yarrowia lipolytica cells modified with phyto-inspired Fe0/Fe3O4 nanoparticles. Journal of contaminant hydrology, 146, pp.63-73.

More specifically, the authors should define the term ‘green synthesis’ and provide some context in terms of earlier works. Since mace is a traditional ingredient, it is relevant to mention other such edible/food-grade sources for nanoparticle synthesis (which is also an advantage of this synthesis strategy). – major point

  1. It is not clear (i) why TEM derived size estimates are different from DLS and (ii) why the two size distributions are remarkably different? Following the authors proposed explanation, it is unexplained why surface binding of biomolecules increases the particles sizes from 5-28 nm to precisely 50 nm (i.e. a size narrowing effect).

  1. Line 93: Provide details of aqueous extract preparation

  1. Line 95: Was sunlight exposure essential in the synthesis? This is a key detail to mention

  1. Line 123: Biomolecules bound to nanoparticles are usually observed as the ‘corona’ or amorphous shell in TEM images. If observed, please discuss.

  1. Line 160-164: Are there bioactive molecules with reducing potential already reported in mace? It be be interesting to mention these here. This could support a mechanism analogous to that of gold NPs synthesized by pomegranate extracts (see figure 6, Facile synthesis of size-tunable gold nanoparticles by pomegranate (Punica granatum) leaf extract: Applications in arsenate sensing. Materials Research Bulletin, 48(3), pp.1166-1173).

  1. Are there antifungal or antimicrobial activities of mace alone? This is an important discussion point. See: "Isolation and characterization of two antimicrobial agents from mace (Myristica fragrans)." Journal of natural products 54, no. 3 (1991): 856-859The antibacterial and antifungal activity of essential oil derived from the flesh of nutmeg fruit. EurAsian Journal of BioSciences, 13(1), 93-98

  1. Line 173: How is the unit ‘%’ defined for antifungal activity?

  1. Figures can be improved: Figure 7 lacks y-axis label. Figure 5 lacks a y-axis.

Author Response

I have revised the article as per the suggestion from the Reviewer and added all the comments in my report.

Reviewer 2 Report

The paper could be published, but only after a major revision

- first, section 3 is missed!

- Quality of presentation is quite low, most of images seems deformed (fig 4,  8 and 11) and/or too big (fig 8 and 11), graphs should be better formatted (for example the one in fig 13 is to o big and with too big lettering...)

- LSPR should be used instead of SPR

- Fig 5, 6 and 7  could be moved in the supplementary material

- An histogram coming from TEM, in order to have an idea of the dispersion of the preparation, should be given, maybe as an inset of figure 4

- Indeed, the discussion of antibacterial effects of  AgNP should be a little more complete. Authors do not mention a mechanism which is strongly supported by several authors, the mechanism involving silver ions release. A couple of recent review should be cited about the debate between the two mechanism:  Int. J. Mol. Sci. 2019, 20, 449 and European Journal of Inorganic Chemistry 2018 (45), 4846-4855. Moreover, this discussion in my opinion should be moved to the introduction and not in the results and discussion part

- going to the antibacterial effects, authors must comment why no effects are observed for P. aeruginosa (and not aeroginosa as used in the paper).

Round 2

Reviewer 2 Report

paper is publishable